

# Evolutionary patterns of range size, abundance and species richness in Amazonian angiosperm trees

Kyle Dexter[1,2] and Jérôme Chave[3]

[1] School of GeoSciences, University of Edinburgh, Edinburgh, United Kingdom
[2] Royal Botanic Garden Edinburgh, Edinburgh, United Kingdom
[3] Laboratoire Evolution et Diversité Biologique, Centre National de la Recherche Scientifique, Toulouse, France

## ABSTRACT

Amazonian tree species vary enormously in their total abundance and range size, while Amazonian tree genera vary greatly in species richness. The drivers of this variation are not well understood. Here, we construct a phylogenetic hypothesis that represents half of Amazonian tree genera in order to contribute to explaining the variation. We find several clear, broad-scale patterns. Firstly, there is significant phylogenetic signal for all three characteristics; closely related genera tend to have similar numbers of species and similar mean range size and abundance. Additionally, the species richness of genera shows a significant, negative relationship with the mean range size and abundance of their constituent species. Our results suggest that phylogenetically correlated intrinsic factors, namely traits of the genera themselves, shape among lineage variation in range size, abundance and species richness. We postulate that tree stature may be one particularly relevant trait. However, other traits may also be relevant, and our study reinforces the need for ambitious compilations of trait data for Amazonian trees. In the meantime, our study shows how large-scale phylogenies can help to elucidate, and contribute to explaining, macroecological and macroevolutionary patterns in hyperdiverse, yet poorly understood regions like the Amazon Basin.

# INTRODUCTION

Some Amazonian tree species attain incredibly high abundance (tens to hundreds of millions of mature individuals), while most have small populations sizes, numbering in the thousands to tens of thousands of individuals (*Ter Steege et al., 2013*; *Ter Steege et al., 2015*). Similarly, the range of some Amazonian tree species extends across the entire Amazon basin, while most are restricted to much smaller areas (*Kristiansen et al., 2009*). A similar imbalance is observed in species to genus ratios. Over half of all Amazonian tree species belong to genera with 100 or more species, while the majority of genera (52%) have ten or fewer species (*Gentry, 1993*). The drivers of variation in these fundamental characteristics of tree species and genera are poorly known, despite the Amazon representing the largest tropical forest in the world and housing the greatest number of tree species of any forest (*Fine & Ree, 2006*).

Corresponding author
Kyle Dexter, kyle.dexter@ed.ac.uk

Variation in the abundance of tree species at the community level is evident in any forest and has been studied in tropical forests since at least 1979 (*Hubbell, 1979*). However, only relatively recently has sufficient taxonomic and forest inventory work been carried out at sufficiently large spatial scales (>1,000 km) to explore patterns of dominance across the Amazon; the results have shown conclusively that certain tree species dominate at large spatial scales as well (*Pitman et al., 2001*; *Ter Steege et al., 2013*; *Ter Steege et al., 2015*). Neutral ecological models where species drift randomly in abundance over time can generate variation in abundance, but they fail to explain the consistent dominance of some tree species across the Amazon (*Hubbell, 2001*; *Condit et al., 2002*). This begs the question of which characteristics or traits allow certain tree species to dominate, but the absence of adequate trait datasets has limited attempts to answer this question. There are two large-scale databases available that provide reasonable coverage of Amazonian tree species, for wood density and seed mass (*Zanne et al., 2009*; *Royal Botanic Gardens Kew, 2016*), but these two traits do not appear to be related to tree abundance in the Amazon (*Ter Steege et al., 2013*; *Ter Steege et al., 2015*). At smaller spatial scales, tree height has been shown to be positively correlated with abundance (*Pitman et al., 2001*; *Arellano et al., 2015*), but it is unclear if this pattern holds at the scale of the entire Amazon Basin.

There has been less progress in studying variation in the range size of Amazonian trees, in large part because many areas of the Amazon remain poorly known by botanical scientists. In 1999, Pitman and colleagues noted that "not a single tree species distribution in the Amazon basin has been reliably mapped" (*Pitman et al., 1999*), and this remains the case. Nevertheless, with some simplifying assumptions, *Feeley & Silman (2009)* succeeded in providing a first estimate of the range size of thousands of Amazonian plant species. These authors documented substantial variation in the range size of Amazonian tree species, but, given their focus on conservation, they did not attempt to explain this variation, and neither has any subsequent study. Range size has been shown to be positively related to the total abundance of tree species in the Amazon (*Ter Steege et al., 2013*), but the arrow of causality probably goes in the reverse direction (i.e., tree species with larger ranges can achieve greater total abundance). Meanwhile, studies of a more limited taxonomic scope have shown that range size in Amazonian palm tree species (Arecaceae) is positively related to their height (*Ruokolainen et al., 2002*; *Kristiansen et al., 2009*). Whether this pattern holds for dicotyledenous Amazonian trees is unknown, and whether other tree traits are related to range size remains to be explored.

Lastly, while variation in the species richness of tree genera in the Amazon has been noted (e.g., *Bermingham & Dick, 2001*), it has received surprisingly little research attention compared to the extensive efforts directed toward understanding the extraordinary *total* tree species richness of the Amazon (*Prance, 1982*; *Hoorn et al., 2010*). *Baker et al. (2014)* represents a landmark in this regard, as the authors focused on explaining variation in species richness *amongst* 51 Amazonian tree lineages (primarily genera). The authors focused on assessing the role of intrinsic factors (i.e., traits of the lineages themselves) as compared to extrinsic factors (e.g., geological events) in explaining this variation. They showed that the species richness of genera was negatively related with their turnover time; genera that showed higher mortality and recruitment rates in forest inventory plots also

had more species. Meanwhile, they did not find a relationship of species richness with dispersal syndrome, breeding system or tree height. The negative results in this study may be best considered as preliminary however, given that the study covered only 51 tree genera (among 100s in the Amazon) and only trees $\geq$10 cm diameter at breast height (1.3 m above the ground), which was the minimum size threshold for sampling trees in the forest inventory plots used to estimate turnover times.

Acquiring sufficient trait data to thoroughly evaluate the drivers of variation in the abundance, range size and species richness of Amazonian tree lineages will require many costly field campaigns that will likely take multiple decades to complete. In the meantime, alternative approaches should be pursued. One fundamental question is whether intrinsic or extrinsic factors are more important. To the degree that intrinsic factors or traits (e.g., tree height or dispersal syndrome) show phylogenetic signal (cf. *Freckleton, Harvey & Pagel, 2002*; *Blomberg, Garland & Ives, 2003*), large-scale phylogenies could be used to test the role of intrinsic factors in driving variation in abundance, range size and species richness. If closely related lineages tend to have similar abundance, range size and species richness, then phylogenetically correlated traits are likely to be important drivers of variation in these characteristics. It is improbable that extrinsic factors would generate phylogenetic signal for abundance, range size or species richness within a given biogeographic region. Recent advances in the generation of sequence data for large numbers of Amazonian tree genera (e.g., *Baraloto et al., 2012*) mean that phylogenetic approaches are now feasible. Here, we generate a temporally-calibrated phylogenetic hypothesis that includes half of all Amazonian tree genera. We use this genus-level phylogeny to assess if there is phylogenetic signal for the mean range size, mean abundance and species richness of genera, with the aim of testing the importance of intrinsic traits of genera in driving macroecological and macroevolutionary patterns in Amazonian trees.

## METHODS

We intersected a list of all Neotropical tree genera (from http://ctfs.si.edu/webatlas/neotropicaltree/) with a list of Amazonian plant species (*Feeley & Silman, 2009*) in order to generate a list of Amazonian tree species. The *Feeley & Silman (2009)* dataset additionally includes estimates of range size for all species. We obtained estimates for the total abundance of Amazonian tree species from *Ter Steege et al. (2013)*.

We obtained sequences of the *rbcL* plastid gene for 631 Amazonian angiosperm tree genera (Table S1), with 567 sequences coming from Genbank (www.ncbi.nlm.nih.gov/genbank/) and an additional 64 genera being newly sequenced following protocols outlined in *Baraloto et al. (2012)*. We obtained sequences of the *matK* plastid gene from Genbank for 452 of the 631 genera with *rbcL* data (Table S1). Sequences were aligned using the MAFFT software (*Katoh & Standley, 2013*). 'Ragged ends' of the sequences that were missing data for most genera were manually deleted from the alignment. Preliminary phylogenetic analyses allowed us to exclude sequences from GenBank for genera that were phylogenetically placed in a different family to that which they are thought to belong taxonomically. The final alignment can be found in Supplemental Information.

We estimated a maximum likelihood phylogeny for the genera in RAxML v8.0.0 (*Stamatakis, Hoover & Rougemont, 2008*), on the CIPRES web server (www.phylo.org). We used the default settings, including a General Time Reversible (GTR) + Gamma (G) model of sequence evolution, with separate models for the *rbcL* and *matK* genes (i.e., a partitioned analysis). We included sequences of *Amborella trichopoda* (Amborellaceae) and *Nymphaea alba* (Nymphaeaceae) as outgroups. This phylogeny (see Supplemental Information) was used as a starting tree for simultaneous topology and divergence time estimation in the software BEAST v1.82 (*Drummond & Rambaut, 2007*). We implemented fossil-based age constraints for 25 nodes distributed across the phylogeny, using log-normal prior distributions with an offset to impose a hard minimum age (see Table S2). We used a GTR + G model of sequence evolution, with separate models for the *rbcL* and *matK* genes, an uncorrelated relaxed lognormal clock, and a birth-death model for the speciation process. We conducted several preliminary runs to optimise the tuning and weight of parameters as per recommendations generated by the software. Once parameter optimisation stabilised, we ran two separate chains for 100 million generations. The first 50 million generations of each chain were discarded as "burn-in," as the posterior probability of the phylogeny did not stabilise until this point. We combined the post burn-in posterior distributions of parameters and confirmed that effective sample size (ESS) values exceeded 100 for all parameters. We then used the phyutility software (*Smith & Dunn, 2008*) to generate an all-compatible consensus tree from the combined post burn-in posterior distribution of trees. Node ages were optimised onto this consensus phylogeny as the median value for a given node across all trees in the posterior distribution that contained the node (using the TreeAnnotator software, *Drummond & Rambaut, 2007*).

For each genus in the phylogeny, we calculated the mean range size and abundance for all constituent species in the *Feeley & Silman (2009)* and *Ter Steege et al. (2013)* datasets. Of the 631 genera in the phylogeny, 493 had an abundance estimate for at least one species in *Ter Steege et al. (2013)*. We considered the number of species for each genus in the *Feeley & Silman (2009)* dataset as an estimate of the species richness of that genus in the Amazon. As an alternative estimate, we used the Neotropical species richness estimates for genera in *Gentry (1993)*, which produced highly similar results. We assessed correlations amongst these genus-level characteristics using Pearson's correlation coefficients for both the raw values and for their phylogenetically independent contrasts.

We tested for phylogenetic signal for each of these genus-level characteristics using Pagel's $\lambda$ (*Freckleton, Harvey & Pagel, 2002*). Under Brownian motion evolution, where trait values drift randomly over evolutionary time and where the phylogenetic relationships of taxa perfectly predict the covariance among taxa for trait values, the expected value of $\lambda$ is one. When the phylogenetic relationships of taxa do not predict the covariance at all, the expected value of $\lambda$ is zero. We compared the fit of different values for $\lambda$ (one, zero and the maximum likelihood estimate) using the Akaike information criterion (AIC).

In order to determine which lineages may be responsible for significant phylogenetic signal for a given characteristic (e.g., mean range size of genera), we used the following approach. We first estimated the ancestral value at each node in the phylogeny using maximum likelihood ancestral state reconstruction (*Schluter et al., 1997*). We then

randomised the tips of the phylogeny 1,000 times, reconstructed ancestral values at nodes each time, and compared the observed reconstructed value to that across the randomisations. If the observed value for a node was greater than that in 97.5% of the randomisations, we considered the lineage descending from that node to show significantly high values for the trait, while if the observed value was lower than 2.5% of the randomisations, we considered the lineage to show significantly low values.

In order to assess whether major clades (Magnoliids, Monocots, Rosids and Asterids) differ in the species richness and mean range size and abundance of their constituent genera, we used analyses of variance with major clade as the grouping variable. In order to determine which clades may be driving significant results, we used Tukey's tests. All analyses were conducted, and figures constructed, in the R Statistical Software (*R Core Development Team, 2016*) using functions in the ape (*Paradis, 2016*), geiger (*Harmon et al., 2016*) and phytools (*Revell, 2016*) packages (see Supplemental Information for codes).

## RESULTS

The phylogeny derived from the DNA sequence dataset that we compiled spans from the Magnoliids to the Asterids, thus encompassing all major clades of angiosperms (Fig. 1; see Supplemental Information for newick-formatted tree file). The 64 *rbcL* sequences that were generated as part of this study represent ∼10% of the genera in the phylogeny, and they therefore represent a valuable contribution to our understanding of the evolutionary relationships of Amazonian tree genera. Most orders and families were monophyletic in the phylogeny with the notable (previously known) exceptions of Olacaceae and Icacinaceae, while the large-scale phylogenetic relationships are largely in agreement with those from recent, angiosperm-wide phylogenetic analyses (e.g., *Magallón et al., 2015*).

The species richness of genera is negatively correlated with mean range size ($r = -0.40$, $p < 0.001$) and mean abundance ($r = -0.38$, $p < 0.001$). These relationships are weaker, but still significant, when using phylogenetically independent contrasts (PICs), indicating that phylogenetically related traits partially underlie the correlations (mean range size PICs: $r = -0.28$, $p < 0.001$; mean abundance PICs: $r = -0.24$, $p < 0.001$; Fig. 2). Meanwhile, mean range size and abundance of genera are strongly positively correlated, using both the raw data and PICs ($r = 0.44$, $p < 0.001$; PICs: $r = 0.43$, $p < 0.001$). All of the genus-level characteristics show significant phylogenetic signal, but less than would be expected under a Brownian motion model of evolution (Table 1).

Significant phylogenetic signal for these characteristics is driven by significantly high or low values in many lineages (Fig. 1, Table S3). Diverse lineages in the Magnoliids and the Asterids show high species richness and low mean range size and abundance, including the Lamiales and multiple lineages in the Rubiaceae and Solanales. One marked exception to the general pattern in the Asterids is Lecythidaceae, which shows low species richness and high abundance. Meanwhile, many lineages in the Rosids show low species richness and high mean range size and abundance, including Euphorbiaceae, Salicaceae and Moraceae. Within the Rosids, the exception to this pattern is found in multiple lineages in the Myrtales, including the Melastomataceae, which show a pattern similar to most lineages in

A) $\log_{10}$(Species richness)    B) $\log_{10}$(Mean range size, km$^2$)    C) $\log_{10}$(Mean abundance, # Inds)

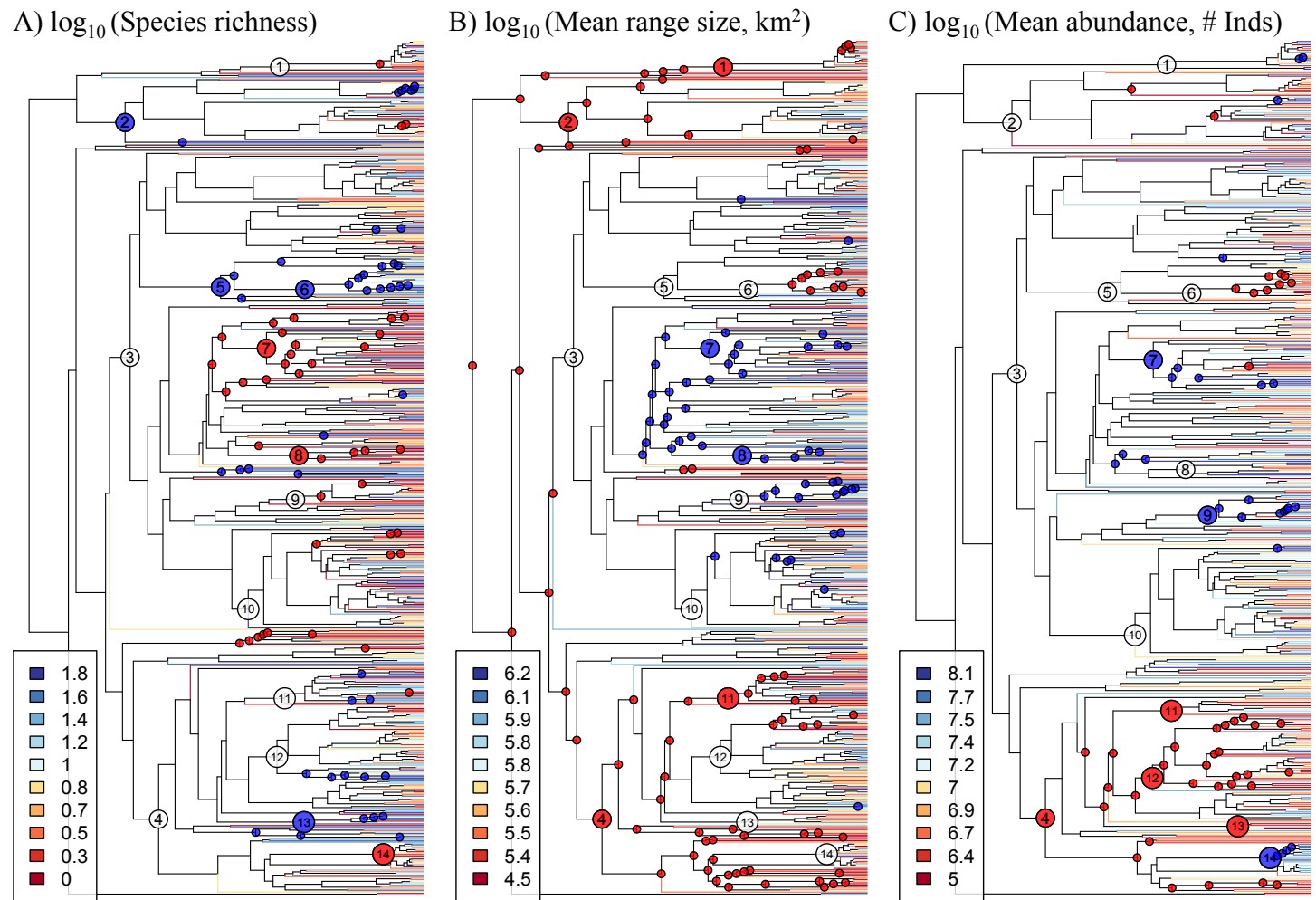

**Figure 1 Phylogeny of 631 Amazonian tree genera with terminal branches coloured according to the (A) species richness, (B) mean range size, and (C) mean abundance of genera.** The following numbered nodes are mentioned in the main text: 1, Arecaceae; 2, Magnoliids; 3, Rosids; 4, Asterids; 5, Myrtales; 6, Melastomataceae; 7, Euphorbiaceae; 8, Salicaceae; 9, Moraceae; 10, Leguminosae; 11, Lamiales; 12, Rubiaceae; 13, Solanales; and 14, Lecythidaceae. Nodes that are coloured blue indicate lineages whose constituent genera show significantly higher values for the given genus-level characteristic than expected by chance, while nodes coloured red show significantly lower values than expected by chance.

the Asterids. The Leguminosae (Fabaceae), the most genus-rich family in our dataset, does not show any significant departures from null expectations, although individual lineages therein show low species richness and high mean range size. Within the monocots, the Arecaceae show low mean range size, while one lineage (*Iriartea* with *Socratea*) shows particularly high abundance.

A non-phylogenetic comparison of genera in the major clades shows they are not significantly different in terms of mean species richness (ANOVA for species richness: $F = 1.18$, $p = 0.317$), while they are significantly different for mean range size and abundance, but the effect sizes are small (range size: $F = 10.56$, $p < 0.001$, $R^2 = 0.05$; abundance: $F = 7.13$, $p < 0.001$, $R^2 = 0.04$). Tukey's tests reveal that genera in the Rosids have significantly larger mean ranges, on average, than genera in the Asterids ($p < 0.001$)

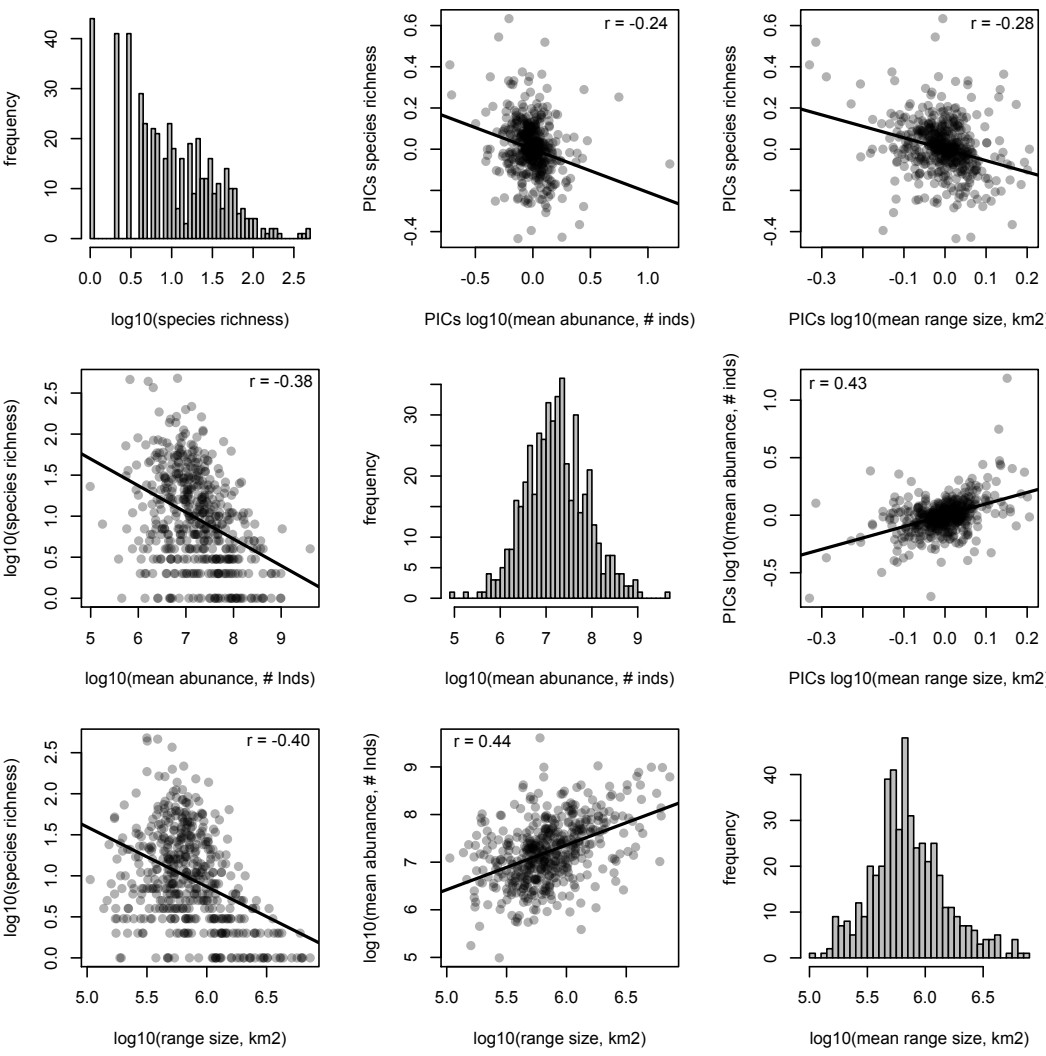

**Figure 2** **Relationships between species richness and mean range size and abundance for Amazonian tree genera.** Histograms of each genus-level characteristic are given on the diagonal. Below the diagonal the raw relationships are shown along with the best-fit linear relationship and the Pearson correlation coefficient. Above the diagonal, the relationships of phylogenetically independent contrasts (PICs) are shown along with the best-fit linear relationship that is forced through the origin and the Pearson correlation coefficient.

and Monocots ($p = 0.019$), while genera in the Rosids and Monocots have significantly higher mean abundance than genera in the Asterids ($p < 0.001$ & $p = 0.020$, respectively).

## DISCUSSION

Fundamental characteristics of Amazonian tree genera, namely their species richness and the mean range size and abundance of their constituent species, show marked and significant phylogenetic signal (Table 1). In other words, closely related genera tend to have similar numbers of species and species with similar range sizes and abundances (Fig. 1). These genus-level characteristics are also strongly correlated with each other (Fig. 2). Our
**Table 1  ΔAIC values for different evolutionary models of trait evolution for genus-level characteristics and for different values of Pagel's λ.**

| Genus characteristic | Evolutionary model | λ | ΔAIC |
|---|---|---|---|
| Log (Species richness) | Estimated λ | 0.26 | 0 |
| | No phylogenetic dependence | 0 | −17.1 |
| | Brownian motion | 1 | −287.8 |
| Log (Mean range size) | Estimated λ | 0.37 | 0 |
| | No phylogenetic dependence | 0 | −52.0 |
| | Brownian motion | 1 | −257.9 |
| Log (Mean abundance) | Estimated λ | 0.32 | 0 |
| | No phylogenetic dependence | 0 | −34.2 |
| | Brownian motion | 1 | −287.5 |

results suggest that intrinsic factors (i.e., traits of the genera themselves) exert a strong influence on the range size, abundance and species richness of Amazonian tree species and genera.

We propose that tree height may be one of the key traits underlying the observed results. Many of the lineages in our study that show high species richness and small geographic ranges (e.g., Myrtaceae, Melastomataceae, Rubiaceae, Asterales, Solanales, and Lamiales) tend to be small in stature. Previous studies have shown a positive relationship between the height of Amazonian trees and their range size (*Ruokolainen et al., 2002*; *Kristiansen et al., 2009*). Such a relationship may be due to larger-statured trees being able to disperse their seeds greater distances, likely through greater fecundity, which would increase the chances that at least some seeds make it a long distance and would, for animal-dispersed species, potentially attract more dispersers. Increased dispersal ability would also increase gene flow among distant populations, which, in turn, could reduce opportunities for allopatric isolation and contribute to reduced diversification. Smaller statured trees may also have shorter generation times, which could contribute to increased diversification (*Baker et al., 2014*). Thus, small-stature may be a biological trait that spurs diversification and may also underlie the negative correlation between mean range size and species richness of genera. Small-statured lineages also show lower abundances in the datasets assembled, although this is partly, if not entirely, explained by the abundance estimates being derived from tree plots that survey individuals >10 cm diameter at breast height (*Ter Steege et al., 2013*). Meanwhile, the relationship between mean abundance and species richness of genera may simply be a by-product of correlations between range size and abundance. These are likely necessarily related as the range size of a species will restrict the total abundance that it can achieve.

Our focus on tree height as a key variable does not negate a role for other phylogenetically correlated traits in contributing to the observed results. In local-scale studies of Amazonian tree communities, most of the traits that have been examined (e.g., wood density, specific leaf area) have shown significant phylogenetic signal (*Kraft & Ackerly, 2010*; *Baraloto et al., 2012*). Large-scale compilations of trait data from across the Amazon are now needed in order to assess which exact traits may be driving our results.

It is important to note that the phylogenetic signal we observe for these genus-level characteristics is less than would be expected under a Brownian motion model of evolution. In fact, the Brownian model fits the data much worse than a simple model whereby there is no phylogenetic dependence to the observed values (Table 1). This result may be due to some evolutionary process, such as selection, driving closely related genera to diverge in values for the studied genus-level characteristics. However, in contrast to what might be thought for other traits (e.g., those related to ecological niches such as resource acquisition or defence strategy), it is not immediately clear why selection would favour divergence among closely related lineages in their diversification rate, average range size or average abundance. Alternatively, a Brownian motion model may be simply a poor descriptor of how these genus-level characteristics change over time. The Brownian motion model stipulates that per unit time, small changes in trait values are much more likely than large changes. It may be the case that large changes in range size, abundance and diversification rate among closely related changes are just as likely as small changes. The exact pattern of change in these characteristics would depend on how the underlying driving factors themselves, be they intrinsic or extrinsic, change over time and space.

In any case, the phylogenetic signal we document is evident across multiple phylogenetic scales. At a broad scale, we find that various lineages in the Rosids are comprised of genera that show low species richness and high mean range size and abundance, while lineages in the Magnoliids and Asterids show the opposite pattern. Interestingly, a non-phylogenetic approach, using analyses of variance, did not find large differences amongst these major lineages. This contrast in results may be due in part to lineages within the major clades that belie the general pattern. For example, the Myrtales (in the Rosids) show a pattern typical of the Asterids, while the Lecythidaceae (in the Asterids) show a pattern typical of Rosids. The contrast is likely also due to the manner in which these genus-level characteristics are distributed across the phylogeny. While genera within a given major clade can show a diversity of values for these characteristics, their relative phylogenetic positions result in reconstructed ancestral values that show systematic differences amongst lineages in different major clades. A phylogenetic approach was thus necessary to reveal how these major clades differ have differed over evolutionary time. This phylogenetic approach has also served to show how phylogenetically-correlated factors, intrinsic to lineages themselves, have contributed to the macroecological and macroevolutionary patterns observed in present-day Amazonian trees.

## ACKNOWLEDGEMENTS

We wish to thank Julien Vieu, Mailyn Gonzalez, Darin Penneys, and Céline Vicédo for generating DNA sequences, and Lourens Poorter, and Tim Baker for collecting new samples in the field. We also thank two anonymous reviewers for helpful suggestions that improved the manuscript.

## Funding

Kyle Dexter was supported by a fellowship from the Centre National de la Recherche Scientifique during the time this research was conducted. Financial support received from Fondation pour la Recherche sur la Biodiversité, and from Investissement d'Avenir grants of the ANR (CEBA: ANR-10-LABX-0025; TULIP: ANR-10-LABX-0041). The funders had no role in study design, data collection and analysis, decision to publish, or preparation of the manuscript.

## Grant Disclosures

The following grant information was disclosed by the authors:
The Centre National de la Recherche Scientifique.
Fondation pour la Recherche sur la Biodiversité.
Investissement d'Avenir grants.

## Competing Interests

The authors declare there are no competing interests.

## Author Contributions

- Kyle Dexter conceived and designed the experiments, performed the experiments, analyzed the data, contributed reagents/materials/analysis tools, wrote the paper, prepared figures and/or tables, reviewed drafts of the paper.
- Jérôme Chave conceived and designed the experiments, performed the experiments, wrote the paper, reviewed drafts of the paper.

## DNA Deposition

The following information was supplied regarding the deposition of DNA sequences:
We used ∼1000 sequences already available on Genbank which are listed in Table S1. We generated 64 new sequences: Genbank KX640832–KX640895.

## Data Availability

The phylogeny that forms the basis of this paper is available at TreeBase: https://treebase.org/treebase-web/search/study/summary.html?id=19277.

## Supplemental Information

Supplemental information for this article can be found online at http://dx.doi.org/10.7717/peerj.2402#supplemental-information.

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
