# Peer review of "Evolutionary patterns of range size, abundance and species richness in Amazonian angiosperm trees"

_PeerJ, doi:10.7717/peerj.2402_

## Round 0.1 · original submission · Major Revisions

· Academic Editor

Major Revisions

Two referees provided extensive review of this manuscript. Even though they agreed that this is a valuable contribution to a timely topic, they also raised a number of points that have to be addressed in a revised version. In particular they both felt that the theoretical background should be better introduced and explained, as the hypothesis tested. Both referees also requested that all data and R scripts should be fully made available. In addition to this, the two referees also spotted a number of issues that should be fixed in the revision.

Reviewer 1 ·

Basic reporting

The manuscript is a brief report of few, very specific results. The variables evaluated are quite fundamental and broadly known. Therefore, it is not really necessary an extensive Introduction. However, the current Introduction is so brief that the reader cannot track relevant publications to see whether other people have worked on these ideas or not. More background on hypotheses and theory would be appropriate.

The work is self-contained but seems to be a small slice of a broader project. For example, there could be interesting results relating the age of the genera and the evaluated properties. Older genera have had more time to speciate, or may contain older species which have had more time to expand their ranges. Another example: the authors cite the speciation rate as one potential mechanism driving their results, but they don't employ the speciation rate in any analysis. Finally, they cite individuals' stature as a potential driver of the observed patterns, but no attempt to include any trait in the analyses is made. It is evident that the authors have access to relevant data about these potential drivers. It is unclear why such data and analyses are not addressed here.

Figures are appropriate and of sufficient resolution. I would say that a rank-abundance plot, and similar plots for range size and genus richness, are worth to include.

Funding agencies should not be listed in Acknowledgements.

The phylogenetic tree has been made available. Unfortunately I haven't been able to open it from R (I have tried read.nexus on the .nex file, and read.tree and get_trees(nexml_read(x)) on the .xml file).

The raw data on abundances and distribution ranges are not available. The authors use data from other studies but if those data are open-access or not it is unclear. Relevant links to the original datasets are not provided. In general, if someone wants to reproduce these results from scratch, he/she should invest a good time finding the raw data.

Experimental design

The manuscript goes to Results almost directly. The submission does not clearly define the research question, nor lists any theory-based prediction regarding the evaluated patterns. The knowledge gap being investigated is not clearly identified, nor references provided so the reader can really check if there is a gap or not. There are no statements as to how the study contributes to filling that gap. In general, the authors provide interesting results but these are provided without sufficient context. For readers familiar with these ideas it is a good thing (short and direct work). For other readers, however, it will be difficult to figure out the relevance of these work and how to find related literature with which contrast this.

The investigation has been conducted rigorously and to a high technical standard. Methods are sufficiently described. The analyses should be easily reproducible by any other investigator familiar with the methods.

Validity of the findings

To my knowledge, the employed data are robust, statistically sound, and controlled. Datasets are among the most suitable to address the topic explored.

First paragraph of the Discussion is a "zoom" into some of the Results. Second and third paragraphs of the Discussion are speculations based on a very subjective and general perception of some clades being shorter (in stature) than others. Although it may well be the case for the clades cited (and I largely agree with the authors), this works does not present any data supporting that distinction between short and tall taxa. I encourage the authors to include data on species traits if >50% of the Discussion is based on tree height and/or other aspects related or correlated with adult tree size.

Additional comments

The results are novel are interesting, and should have an important impact on how we understand the evolution of tropical species ecologies. However, impact, degree of advance, novelty, etc, are not among the review criteria for this journal. Thus, I'm afraid that I cannot provide a positive review on your work. The most important reason is that this work appears to be a tiny piece of a much broader and ambitious project. There are several aspects relevant to the ideas presented here that (apparently) have been purposely omitted, like clade age, clade speciation rates, or even traits. I appreciate the direct and synthetic approach to the writing, but I am not in favour of breaking down good ideas and analyses into too small pieces.

Reviewer 2 ·

Basic reporting

In this study by Dexter and Chave, the relationships between range size, species abundance and species richness are explored using comparative phylogenetic techniques and a mix of large public data and additional genetic data sequenced by this study. The paper is generally clear and concise in it's writing style and the figures presented are sufficient to explain the majority of the results, and are well-formatted.

The introduction is quite brief (given the breadth of this question and amount of potential background), but is acceptable, given that the paper is shorter. I would recommend moving the background and citations about stature to the introduction (but see comments below about the inclusion of extended discussion about stature in the absence of stature data in this study). This move would strengthen the introduction and avoid introducing the stature background a little too late in the paper. Additionally, it would remove a lot of citations from the discussion, which can be distracting and weaken the flow of discussing the author's results.

Reference are generally good, except that the work depends heavily on ter Steege et al 2013, but makes no reference to the more recent and expansive ter Steege et al 2015. This work should at least be cited in the introduction.

Overall, I was unsure if the paper's highlighted results matched the data and analyses that was generated specifically for this study. The sequence data collected, while valuable, seemed to do little to enhance inference of the higher level relationships, and results from taxa specifically sequenced for this study were not highlighted in the results. Also, the connection to the stature data was not fully explained or appropriated supported by data, which made its inclusion in the abstract and discussion confusing. I would say this paper could be a useful contribution, but only after significant rewriting of the manuscript, full detail methods to make the analyses replicable, and consideration of the concerns about the data interpretation described below.

Experimental design

The exact hypothesis to be tested was unclear from both the abstract and the introduction. The authors state in the introduction that the relationships will be "explored" between these variables, but a statement of a specific hypothesis derived from the literature would greatly strengthen the introduction.

For replication, I recommend both the MAFFT-generated and manually edited alignments should both be included in the supplied data. A more thorough description of why the alignment was manually edited would also strengthen the methods. Ideally, the analysis (or at least the tree) should be tried using both alignments to demonstrate that manual editing does not substantively alter the results.

"Preliminary phylogenetic analyses allowed us to exclude sequences from GenBank that likely represent taxonomic misidentifications." Here again, this is not fully described and thus non-replicable. Please add details as to what sequences were excluded and why. At minimum, note these sequences in Table S1. Were these placed completely in the wrong family? Were corrections submitted to GenBank for consideration or consultation before they were excluded? And again, did the exclusion of these results substantively change the results?

No parameters are listed for RAxML or BEAST. What models were used and what parameters? Please describe more fully to allow replication and reanalysis. Including the exact commands used with the all flags and parameter values would be most efficient to allow full replication.

The software used to calculate linear regressions was omitted. From Figure 2, I would guess R, and if so, would strongly recommend including the R scripts.

The software used to plot both Figures was omitted from the methods.

Validity of the findings

The conclusion that range and abundance are correlated is not surprising since range for trees (especially for Amazonian canopy trees) would be expected to be tightly correlated to range size. I would recommend discussing/downplaying this result more directly in the results. While I don't believe this was the intention, the paper as currently written implies that the two variables are independently phylogenetically correlated, and that this is a significant finding of the paper. Again, it does not seem like this was the intention, but I would even potentially recommend avoiding reporting both of these variables in the Tables and Figures since they are tightly linked. An alternative would be to note the strong correlation between these variables in the Methods, then only report results for one of them in the Results and Discussion (or treating them as a composite range-abundance variable).

ll.130-143: This paragraph is a bit repetitive and could be revised to be more concise. The paragraph also frequently uses the word "significant" without statistical support values.

ll.155-157: The broad scale finding mentioned here is only vaguely described in the results. A more concise quantitative summary (like a box plot of the two groups with t-test) would be recommended to support this conclusion. The result is still worth reporting even if the t-test is non-significant, as long as a rough pattern is visible and the results are contextualized accordingly.

ll.159-186 These two paragraphs concern the relationship of the plant stature to phylogeny and range, but are completely unsupported by data. Considering this is treated as a main conclusion and makes up most of the discussion, ideally this should also be one main focus of the paper's quantitative analysis. In order to report this result on the same standing (or greater, as it is at present) with the other data-driven results, a phylogenetic independent contrast and other statistical tests should be performed on this as well. It seems like the additional sequence data collected here might be able to add or dispute these previous studies about stature (if stature information is available about the species), so if the authors would prefer to have this as a primary goal of the study (something that I think would make a good addition), it would just need data support. In regard to an earlier comment, "stature" could even replace one of the tightly linked range or abundance variables.

ll.183-186: This final conclusion of the paper is that "Should many small-ranged species go extinct, more phylogenetic diversity may be lost than if range size were distributed randomly across the phylogeny as deeper phylogenetic branches would be more likely to be lost." This final statement could be greatly enhanced and vastly increase its impact (and the impact of the study's findings in general) if it was reported in the style of the Purvis paper cited. Most simply, just stating that "if the X species with smallest ranges were lost, then X genera/families/orders would be lost" or "if the X genera with smallest average ranges were lost..." would add more impact and emphasis to the conclusion and better develop/highlight the data-driven findings of this study. Otherwise, this last sentence feels like it has no background (Purvis is not explained in the intro) and is thus underdeveloped as a closing conclusion.

Additionally, I have concerns about the interpretation of the results in this last sentence. The general pattern observed is that species from more species-rich genera have smaller average ranges and that species-rich genera are phylogenetically clustered. So the authors have observed that there are two main types of genera that cluster phylogenetically on the tree: genera with many species each of which has smaller average range size and genera with few species that have larger average range sizes. They assume that, on average, species with smaller ranges are more prone to extinction (presumably stochastic drift-based forces that might preferentially affect small populations or range sizes). Then the conclusion is drawn that phylogenetically diversity at levels higher than species richness will be lost under this model and assuming that *species* from the many-species-small-range genera are more prone to extinction.

However, if extinction events for each species in a genus are phylogenetically independent and there are more species (and thus more species to be lost) in many-species-small-range genera, then the rate of *genus* loss should be similar between many-species-small-range and few-species-large-range genera Furthermore, if the total range of all species in a genus (i.e. the genus range) is similar for both types of genera, then the risk of genus-level or higher loss of phylogenetic diversity is also approximately equal. In other words, the pattern observed may be a trade-off in having many named species with smaller ranges or few larger-range species. This would be expected to fluctuate for a genus over evolutionary time too, with species-rich genera losing species and other congeneric species filling the shared niche space made available by extinction, while large-range species over time will attain allopatry that promotes speciation. If genera are fluctuating in number of species but maintaining a fairly constant genus-level range and abundance, then this would be evidence against phylogenetically-biased loss of diversity at clades above the genus level.

One way to address this in the paper (besides discussion) would be to try the analysis shown in figure 2 with genus-level ranges (instead of averaged species-level ranges). If genus-level range size (not mean species range size) is also phylogenetically clustered, then this would be stronger evidence for the potential for broad-scale loss of diversity (above the genus level). Another way to say this is, do the results hold up if you ignore divisions at the species level? Especially in the world of tropical plants, it is debated that taxonomic sampling and the number of named species can vary greatly among genera and families depending on its history and community of research.

Additionally, what is the justification for assuming that on average smaller range correlated with higher extinction? The range of values in this study was 100k km2 (~French Guyana) to 10M km2 (presumably the entirety of Amazonia), which are all reasonably large ranges. At least some empirical literature citation should be used to justify this assumption.

Several data items needed to assess the results are missing:

- Sequence alignments
- The TREEBASE link provided was not accessible, and both the original ML and time-calibrated Bayesian tree should be provided.

Thanks for providing the genus-level data, this was much appreciated. However, the CSV file was missing newline characters, and was not readable by any software I had. Therefore, I was unable to further look at the data to evaluate some of my thoughts listed above. It's possible that these newline characters were removed automatically during transmission to me. Even though the author noted the data are publicly available, I would still recommend inclusion of the data (and scripts if R was used) used to generate Figure 2 for full replicability.

Additional comments

I would optionally suggest altering the title to reflect the paper concerns Angiosperms, not all trees.

l.145 Recommend moving this lone sentence to a paragraph the Methods.

Recommend change to "P"-value and "P =" instead of "p"-value

The x-axes in left-bottom and center-bottom panels of Figure 2 are mislabeled.

---

## Round 0.2 · Minor Revisions

· Academic Editor

Minor Revisions

Both referees found that the manuscript has been substantially improved now, and I agree with them. Referee 2 still had some points that should be taken into account before final acceptance.

Reviewer 1 ·

Basic reporting

The article has improved since the first submission. The context for the results is now clear. The response of the authors and the changes implemented are fully satisfactory. There is only the issue of the raw data availability. It would be good if the raw data can be made available (but the authors note some copyright issues regarding the original source of the data). A working nexus tree needs to be stored in TreeBase before publication.

Experimental design

The experimental design is OK. I have no additional comments in this second review.

Validity of the findings

The results are valid and valuable. I have no additional comments in this second review.

Reviewer 2 ·

Basic reporting

In general, the manuscript and style are improved throughout. The enhanced introduction is now sufficient for the paper, and the inclusion of more detailed methods and appropriate supplementary data makes the study's methods easier to follow and replicate. The results are more succinct and clear now as well.

The manuscript is much improved, and will make a somewhat modest, but overall stronger contribution to the discussion in this field.

I think that that only "minor revisions" are necessary to see this through to publication. However, I would ask the authors to seriously consider the comments concerning the Discussion from the first round that I draw attention to again here. This concern for me is not a barrier to the manuscript's publication.

Minor notes:

- Clauses beginning with "i.e." and "e.g." should be inside parentheses throughout the document. Also, I find that in the Abstract these "i.e.," phrases weaken the otherwise succinct structure. Recommend rephrasing these clauses in the Abstract either as separate full sentences or remove them and leave the larger explanations for the introduction.

l.64,71 - Consider adding ter Steege 2015 here as well

l.77 "remain unexplored by scientists using standardised Latin names." - Something about this phrasing feels a little impolitic. Recommend changing to "contain vast numbers of species without formal taxonomic names." (or something similar that focuses on species not people)

l.104 "and only trees ≥10 cm diameter at breast height" - 'at breast height' seems colloquial and inexact (meters?), but if it's a standard heuristic measure in this field, then I have no objections

l.159 - double-quotes should be used

Experimental design

Inclusion of the data and scripts is great.

Methods are now easier to follow and for others to replicate or analyze, and no further changes seem needed.

Validity of the findings

Both reviewers in the first round and the editorial comments cautioned the authors that spending so much of the discussion on the topic of stature that is not explored with data in the paper caused a mismatch between results and discussion. The authors have responded to this by saying that there is no stature data. However, in the present manuscript only 7 lines are devoted to discussion of phylogenetic results, 16 lines at the end for the broad ANOVAs, while 26 lines are devoted to discussing "stature". While I sympathize that data in this area is underdeveloped, I will repeat my concern that so much discussion of a topic not supported by data seems misplaced in this manuscript. I would recommend at least balancing the discussion by increasing the amount of space devoted to discussing the actual results of the paper, so that the discussion is not so dominated by a topic that is not previously discussed in the manuscript, nor addressed by the data or analyses conducted for this study.

---

## Round 0.3 · accepted · Accept

· Academic Editor

Accept

All the corrections and changes requested by the referees have now been done. Thank you for this nice piece of work.